# Data Fusion for Estimating High-Resolution Urban Heatwave Air Temperature

Zitong Wen [1,*], Lu Zhuo [1,2], Qin Wang [1], Jiao Wang [1], Ying Liu [1], Sichan Du [1], Ahmed Abdelhalim [1,3] and Dawei Han [1]

[1] Department of Civil Engineering, University of Bristol, Bristol BS8 1TR, UK; lu.zhuo@bristol.ac.uk or zhuol@cardiff.ac.uk (L.Z.); q.wang@bristol.ac.uk (Q.W.); jiao.wang@bristol.ac.uk (J.W.); emily.liu@bristol.ac.uk (Y.L.); sichan.du@bristol.ac.uk (S.D.); ahmed.abdelhalim@bristol.ac.uk or ahmed_abdelhalim@mu.edu.eg (A.A.); d.han@bristol.ac.uk (D.H.)
[2] School of Earth and Environmental Sciences, Cardiff University, Cardiff CF10 3AT, UK
[3] Geology Department, Faculty of Science, Minia University, Minia 61519, Egypt
[*] Correspondence: bq22490@bristol.ac.uk

**Abstract:** High-resolution air temperature data is indispensable for analysing heatwave-related non-accidental mortality. However, the limited number of weather stations in urban areas makes obtaining such data challenging. Multi-source data fusion has been proposed as a countermeasure to tackle such challenges. Satellite products often offered high spatial resolution but suffered from being temporally discontinuous due to weather conditions. The characteristics of the data from reanalysis models were the opposite. However, few studies have explored the fusion of these datasets. This study is the first attempt to integrate satellite and reanalysis datasets by developing a two-step downscaling model to generate hourly air temperature data during heatwaves in London at 1 km resolution. Specifically, MODIS land surface temperature (LST) and other satellite-based local variables, including normalised difference vegetation index (NDVI), normalized difference water index (NDWI), modified normalised difference water index (MNDWI), elevation, surface emissivity, and ERA5-Land hourly air temperature were used. The model employed genetic programming (GP) algorithm to fuse multi-source data and generate statistical models and evaluated using ground measurements from six weather stations. The results showed that our model achieved promising performance with the *RMSE* of 0.335 °C, *R-squared* of 0.949, *MAE* of 1.115 °C, and *NSE* of 0.924. Elevation was indicated to be the most effective explanatory variable. The developed model provided continuous, hourly 1 km estimations and accurately described the temporal and spatial patterns of air temperature in London. Furthermore, it effectively captured the temporal variation of air temperature in urban areas during heatwaves, providing valuable insights for assessing the impact on human health.

**Keywords:** air temperature; data fusion; downscaling; genetic programming; ERA5-Land; MODIS

## 1. Introduction

Due to global warming, progressively more frequent heatwaves have gradually drawn the attention of the academic community. Considering the variations in population ac-climatisation and adaptation across different regions, the definition of heatwaves also varies accordingly [1]. Generally, it is a period of consecutive days when the weather is excessively hotter and drier than normal conditions [2]. In recent decades, the intensity, frequency, and duration of heatwaves have increased [3]. In 2020, the Centre for Research on the Epidemiology of Disasters (CRED) and the United Nations Office for Disaster Risk Reduction (UNDRR) [4] found that heatwaves have sharply increased by 232% from 2000 to 2019 worldwide. The increase in air temperature can escalate the risk of illness and death for vulnerable residents [5]. The urban heat island (UHI) effect increases the air temperature during heatwaves, especially in urban areas. On the other hand, from 1985 to

2017, the population residing in urban areas increased from 41% to 55% [6]. As a result, heatwaves pose significant risks to urban residents, particularly those living in big cities.

Air temperature is a crucial variable in climate models, and it widely serves as a fundamental metric for defining heatwaves [7]. Meteorological stations generally observe it at 2 m above the land surface with high accuracy and temporal continuity. However, the limited number of meteorological stations restricts their ability to present the spatial distribution of air temperature, especially in urban areas [8]. Abnormal mortality analysis during heatwaves at the city or community level often requires high-resolution air temperature data [9]. On the other hand, understanding the evolution mechanism of heatwaves and their relationship with climate change also relies on accurate and temporally continuous data [10]. Many studies used other data sources to analyse air temperature changes in areas of interest to address these challenges. Reanalysis models and satellite products are among the most commonly used data sources. Reanalysis models, such as the land component of the fifth generation of European reanalysis (ERA5-Land), generally offer good temporal continuity (e.g., hourly), but its spatial resolution is too coarse for urban studies (e.g., $0.1° \times 0.1°$). Zou et al. [11] attempted to use ERA5-Land to evaluate air temperature for coastal urban agglomerations. However, their findings indicated that the coarse spatial resolution of the data made it difficult to differentiate between built-up areas and other land covers (e.g., grasslands). Unlike reanalysis data, satellite products usually offer a higher spatial resolution (e.g., 1 km). However, they often suffer from temporal discontinuity caused by weather conditions, such as cloud cover. For example, even though the Landsat 7 satellite is scheduled to revisit an area every 16 days, the effect of cloud cover can lead to a gap in data availability for several months [12]. Therefore, despite the availability of numerous datasets, none of them alone can meet the requirement for high-resolution air temperature monitoring.

Data fusion has been widely used to obtain higher quality and more relevant information from multi-source data [13]. It was first introduced in the 1960s as a mathematical model that combined data from multiple sources to acquire improved data [14]. In the field of atmospheric science, statistical downscaling-based data fusion has already been used by many scholars to obtain high-resolution temperature data. Abunnasr and Mhawej [8] utilised a linear regression model to integrate multiple satellite products, including the datasets of digital elevation model (DEM), normalised difference vegetation index (NDVI), enhanced vegetation index (EVI), and evapotranspiration (ET), to produce a five-year night air temperature trend analysis with 1 km spatial resolution. Although the research achieved promising results with the coefficient of determination (*R-squared*) of 0.895 and root mean square error (*RMSE*) of 0.49 °C, the five-year temporal resolution was too coarse for the demand of most urban-scale studies. To obtain daily maximum air temperature at 1 km resolution, Dos Santos [15] employed machine learning and six satellite products to calibrate a regression model. However, the performance of the regression model was relatively poor, with the *RMSE* of 2.03 °C and *R-squared* of 0.68. For reanalysis-based datasets, the interpolation method is mainly employed to fuse multi-source data. Combining the reanalysis data and ground measurements, both Wakjira et al. [16] and Viggiano et al. [17] used interpolation approaches to downscale the temperature data. However, their spatial resolutions were still relatively coarse, which were $0.05° \times 0.05°$ and 2 km, respectively. Although some studies have attempted to fuse both satellite and reanalysis datasets using statistical downscaling for air temperature retrievals, their resolution and accuracy are relatively poor for urban-scale research. For instance, Karaman and Akyürek [18] employed a downscaling approach that combines five reanalyses and four satellite products to achieve daily mean temperatures at 0.05° resolution, with an *RMSE* of 2.14 °C. Considering the spatial and temporal autocorrelation of the in situ observed air temperature, Zhu et al. [19] proposed a method for air temperature reconstruction based on the multisource data and machine learning technique. However, despite having *MAE* and *RMSE* both below 0.5 K, the temporal resolution is too coarse, only allowing for monthly data estimation. To acquire high-resolution data, Shen et al. [20] first employed deep learning for estimating 0.01° daily

maximum air temperature based on remote sensing and ground station observations with the *RMSE* of 1.996 °C and *R-squared* of 0.986. Zhang et al. [21] integrated eight types of reanalysis and satellite datasets based on machine learning to obtain 1 km daily average air temperature data. Despite providing a dataset with high spatial resolution, it could not well represent temperature changes during urban heatwaves due to its low accuracy with an *RMSE* of 1.70 °C. Zhang et al. [22] further explored the potential of machine learning in fusing multi-source data to obtain high-resolution and accurate air temperature data. Their team developed a novel five-layer deep belief network deep learning model to generate daily air temperature data, yielding promising results with an *RMSE* of 1.086 °C and an *R-squared* of 0.986. However, due to the limitation of the temporal resolution of explanatory variables (daily), this method cannot further estimate hourly air temperature. In the field of public health, the daily temporal resolution remained inadequate for accurately assessing the duration of high-risk periods within a single day during the heatwave. Therefore, it is imperative to explore a relatively simple and highly accurate statistical method for fusing multi-source data to acquire high-resolution air temperature data at the city level.

The Genetic Programming (GP) algorithm is widely used in atmospheric science as a data fusion technique. Since the GP algorithm is an extension of the genetic algorithm, it can automatically generate interpretable statistical climate models by combining multiple data sources based on genetic evaluation [23]. Similar to genetic algorithm, GP algorithm can provide a relatively simple approach to identify optimal solutions without requiring individuals to have extensive knowledge of the specific problems. The research of Stanislawska et al. [24] proved the potential of the GP algorithm for air temperature downscaling (*R-squared* > 0.90). Coulibaly [25] also demonstrated that the GP algorithm was more straightforward and efficient for estimating local-scale daily extreme temperature than other statistical methods. Despite the potential benefits, to our knowledge, few studies used the GP algorithm to build downscaling models for obtaining air temperature data at high spatial-temporal resolutions.

In the current context of increasingly frequent heatwave events, an important challenge for scholars is how to efficiently obtain high spatiotemporal resolution air temperature data by fusing multi-source data, which is crucial for analyzing the impacts of heatwaves. To address the current research gap, our study proposed a novel two-step data fusion model that integrates multi-source data while exploring the key explanatory variables for accurate air temperature estimation. To illustrate the effectiveness of our model, we conducted a case study in London. Specifically, moderate resolution imaging spectroradiometer (MODIS) land surface temperature (LST) and other satellite-based local-scale variables, including NDVI, normalized difference water index (NDWI), modified normalized difference water index (MNDWI), elevation, emissivity, and ERA5-Land hourly air temperature were fused to generate a two-step statistical downscaling model by using GP-assisted regression modelling. The fusion of satellite and reanalysis products for estimating temporally continuous air temperature data at high spatial-temporal resolution (hourly, 1 km), especially for studies related to heatwaves, has not been explored in previous literature. Thus, such datasets can significantly benefit local authorities in assessing heatwave-related health risks, as well as other heatwave-related studies such as resilient urban planning.

## 2. Study Area and Datasets

### 2.1. Study Area

In the UK, heatwave mainly affects London, which is located in the southeast of England. More than 9 million residents live here, with an average density of 5700 residents per square kilometre. From 1960 to 2019, 66 heatwave events have been reported with an increasing trend [26]. Due to the influence of the UHI, the air temperature in London once approached 40 °C during the 2022 heatwaves [27], resulting in 664 deaths which significantly threatened local residents' health. For instance, the proportion of deaths in London during the 2003 heatwave increased by 42% compared to the same period in 2002 [28]. Therefore, this research focuses on the study area of London, particularly the



region between latitudes 51.7° and 51.3°N and longitudes 0.5°W and 0.1°E. This is because it includes the entire London city, edge cities, water bodies, and forested region, which can provide a better understanding of air temperature distribution across other land covers in urban areas, not just built-up areas. The satellite imagery of the study area obtained from Google Maps is presented in Figure 1a, while Figure 1b showcases its land cover obtained from the MODIS product, known as MCD12Q1.

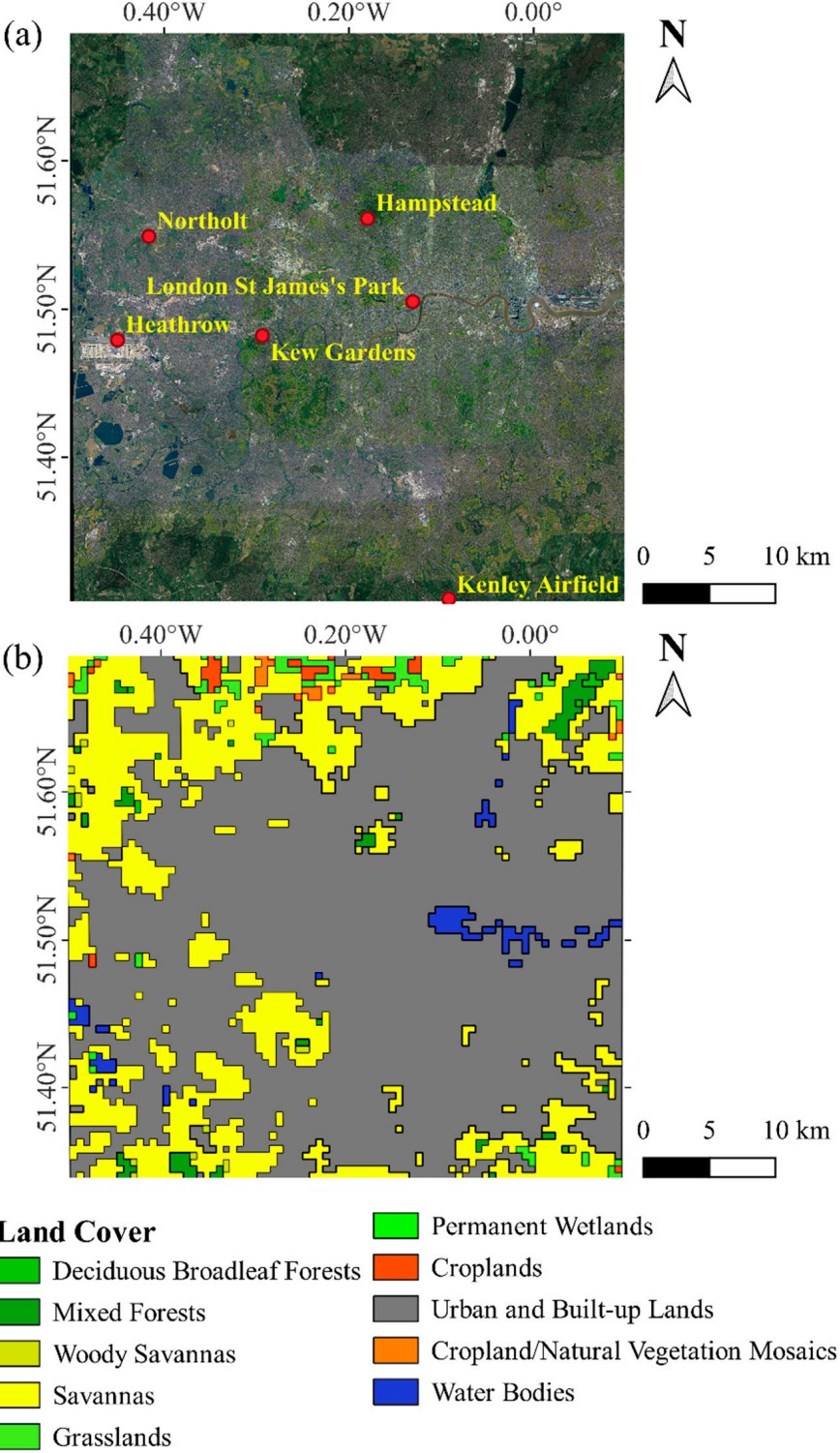

**Figure 1.** Research area: (**a**) satellite imagery with weather stations marked as red dots; (**b**) land cover.

To estimate the temporal variation of air temperature better, five heatwave events that lasted more than three days were selected, as shown in Table 1. All heatwave events were defined using the criteria developed by the Heatwave Plan of England when: (a) a Level 3 heatwave alert has been issued in any part of the country, and/or (b) the mean value of the Central England Temperature (CET) reached 20 °C [29].

**Table 1.** Selected heatwave events between 2011 and 2020.

| No. | Start Date | End Date | Maximum CET (°C) | Source |
|-----|-----------|----------|------------------|--------|
| 1 | 11 July 2013 | 24 July 2013 | 28.1 | [30] |
| 2 | 28 June 2019 | 30 June 2019 | 30.6 | [31] |
| 3 | 21 July 2019 | 28 July 2019 | 34.1 | [31] |
| 4 | 23 August 2019 | 29 August 2019 | 29.9 | [31] |
| 5 | 23 June 2020 | 27 June 2020 | 29.4 | [32] |

### 2.2. Datasets

The products utilised in this study are summarised in Table 2.

**Table 2.** Summary of the data products used.

| Data Type | Source | Product | Temporal Resolution | Spatial Resolution |
|-----------|--------|---------|---------------------|--------------------|
| Temperature products | ERA5-Land | Air temperature | Hourly | 0.1° × 0.1° |
| | MODIS | LST | Daily | 1 km |
| | Open data version of met office integrated data archive system (MIDAS-Open) | Air temperature | Hourly | Point |
| Topographical and geographic products | Shuttle radar topography mission (SRTM) | DEM | / | 90 m |
| | MODIS | Emissivity | 8 days | 1 km |
| | MODIS | NDVI | 16 days | 1 km |
| | Landsat 8 | NDWI | 8 days | 30 m |
| | Landsat 8 | MNDWI | 8 days | 30 m |

#### 2.2.1. Temperature Products

As a reanalysis dataset, ERA5-Land is released by the European Centre for Medium-Range Weather Forecasts (ECMWF). It offers a consistent view of the development of land and atmospheric variables from 1950 to the present. Compared with similar products, such as ERA5 and ERA-Interim, it provides a higher horizontal resolution (0.1° × 0.1°) for hourly information of surface variables. In this study, ERA5-Land provided hourly air temperature data in the downscaling model, which can be accessed at https://cds.climate.copernicus.eu/ (accessed on 2 December 2022).

MODIS on Terra and Aqua are critical instruments used by the Earth Observing System (EOS) program for earth and climate measurements. Its products can observe daily changes of the land surface variables at 1 km resolution over time around the world. Due to its high spatial-temporal resolution, MODIS series products are one of the most used datasets in downscaling-related research. This study selected high-resolution daytime LST data measured by MODIS as a local-scale variable. The MODIS LST products (MOD11A1 and MOD11A1), which provide 1 km daily LST data, were utilised in this study. These data can be freely accessed from Google Earth Engine (GEE): https://doi.org/10.5067/MODIS/MOD11A1.061 (accessed on 2 December 2022) and https://doi.org/10.5067/MODIS/c.061 (accessed on 2 December 2022). However, the data of MODIS products are not always available because of cloud and other weather interferences. Furthermore, due to the satellite motion, Terra and Aqua generally overpass London in the daytime between 10:00 a.m.

and 2:00 p.m. As a result, MODIS products can only provide observational data for the corresponding time.

The observed air temperature data were obtained from the open data version of the met office integrated data archive system (MIDAS-Open), which can be downloadable at http://dx.doi.org/10.5285/3bd7221d4844435dad2fa030f26ab5fd (accessed on 2 December 2022). The data were collected using platinum resistance thermometers at 2 m height in weather stations affiliated with MIDAS-Open. As illustrated in Figure 1a, six weather stations are located across London, including Hampstead (station ID: 695; 51.561°N, 0.18°W), Heathrow (station ID: 708; 51.479°N, 0.451°W), Kew Gardens (station ID: 723; 51.482°N, 0.294°W), Kenley Airfield (station ID: 726; 51.304°N, 0.092°W), London St James's Park (station ID: 697; 51.505°N, 0.131°W), and Northolt (station ID: 709; 51.549°N, 0.417°W). Station-based observation data is widely considered the most reliable data source with the highest resolution and accuracy. To develop and verify the downscaling model, this study utilised hourly temperature data collected from weather stations during the period of 2011–2020. However, as Hampstead station was deactivated in July 2016, only the data from its operating period were utilised while building the model.

### 2.2.2. Topographical and Geographical Products

In addition to the temperature datasets, five topographical and geographical variables, including elevation, emissivity, NDVI, NDWI, and MNDWI, were also adopted in the data fusion model. These variables were regarded as explanatory variables to establish a statistical link with the global-scale variable (the air temperature data from ERA5-Land) so that temporally continuous, high spatial resolution air temperature datasets can be produced. For details, see Section 3.1.

DEM is widely recognised as one of the main factors affecting air temperature, and the typical temperature lapse rate is often regarded as 6.5 °C/km. To quantify its impact in urban areas, the 90 m DEM from shuttle radar topography mission (SRTM) Digital Elevation Data Version 4 was used. It collected over 80% of DEM data around the globe and can be downloaded from https://srtm.csi.cgiar.org (accessed on 2 December 2022).

Emissivity is the ratio of the energy radiated from the surface of a material to that radiated from a blackbody. Due to its strong land-cover dependence, many studies adopted emissivity to determine land types [33]. In this case, it was used as the explanatory variable to quantify the influence of different land covers on air temperature distribution. The MODIS emissivity product known as MOD11A2 was utilised, available at 1 km, 8-day resolution, and can be accessed at https://doi.org/10.5067/MODIS/MOD11A2.061 (accessed on 2 December 2022).

NDVI was utilised to monitor global vegetation conditions and applied for land cover change studies [8]. It can effectively detect the presence of vegetation and estimate vegetation coverage [34]. Thus, to investigate the impact of vegetation on air temperature, we utilised the MODIS NDVI product (MOD13A2) in this case. It can provide 1 km NDVI data with the 16-day temporal resolution, available at https://doi.org/10.5067/MODIS/MOD13A2.061 (accessed on 2 December 2022).

The water body is also one of the main factors that can have a cooling effect on air temperature [35]. Both NDWI and MNDWI served as water indicators in previous research and could accurately identify the presence or absence of water body areas [36]. However, NDWI failed to distinguish between water bodies and built-up areas, while MNDWI tended to classify agricultural wetlands as water bodies due to its high sensitivity to water. Therefore, to more accurately quantify the impact of water bodies on air temperature, both indicators were utilised in the modelling process to complement each other. The Landsat 8 product, known as USGS Landsat 8 Level 2 product, was used to provide 30 m NDWI and MNDWI data with 8-day temporal resolution in this research, available at https://developers.google.com/earth-engine/datasets/catalog/LANDSAT_LC08_C02_T1_L2 (accessed on 2 December 2022).

*2.3. Data Pre-Processing*

Pre-processing steps are needed as the datasets used in this study vary in spatial and temporal resolution. For the spatial resolution of all datasets, it needed to be converted to 1 km resolution in GEE. Expressly, the conversion set the 1 km resolution of the MODIS product as the target and spatially aligned to the same pixel boundaries. High-resolution products, such as Landsat 8 and SRTM, were aggregated to 1 km, and coarse-resolution products, such as ERA5-Land, were reprojected to 1 km. Weather station data were assumed to represent the air temperature of the 1 km pixel where it was located.

For the pre-processing of temporal resolution, the hourly data corresponding to MODIS satellites overpassing time for temperature products were collected. For topographical and geographical products, previous research has widely considered that these variables can be assumed static over a period of time [37,38]. As such, monthly-scale averages of topographical and geographical variables, which were extracted via GEE, were adopted in this study to represent local-scale climate characteristics.

## 3. Two-Step Data Fusion Model

*3.1. Overall Framework*

The proposed statistical downscaling model consists of two main steps, as shown in Figure 2. Scholars widely acknowledge that LST is one of the primary explanatory variables for estimating air temperature [39,40]. Many previous studies have demonstrated strong linear correlations (*R-squared* > 0.80) between LST and air temperature [41–43]. Based on this understanding, the first step involved integrating the air temperature data from ERA5-Land and the LST data from MODIS. This fusion process aimed to generate a statistical downscaling model specifically calibrated to the satellite overpassing time. This step assumes that the obtained downscaling model is applicable to all pixels within the study area, enabling the derivation of high-resolution (1-km) daytime air temperature data within the research area. To ensure an adequate sample size, this study selected data from the Aqua and Terra satellites, obtained during cloud-free conditions, for the summer months (June to August) between 2011 and 2020 to train regression models for downscaling (regression model A). Specifically, the data were collected from 1 km pixels corresponding to six meteorological stations, resulting in a total of 2764 samples.

Since regression model A only worked at satellite overpassing time, to produce temporally continuous datasets, the second step of modelling was carried out. In the second step, it first calculated the differences between the 1 km air temperature data obtained from regression model A and the air temperature data from ERA5-Land at the corresponding time. Then, to explain the differences, regression model B was built by fusing five topographical and geographic variables, including elevation, emissivity, NDVI, NDWI and MNDWI. In this case, under cloud-free conditions, the differences and corresponding explanatory variables during the heatwave events listed in Table 1 were collected within the study area to train the regression model for estimating the differences. Specifically, within the study area comprising 2208 1 km pixels, a total of 63,579 samples were collected for training and validating the regression model. Although the regression model can only explain the differences between 10:00 a.m. and 2:00 p.m. (satellites overpassing time), the atmospheric circulation during heatwave periods is typically controlled by persistent anticyclones, leading to stable weather conditions characterised by cloudless skies and advection of hot air [44]. Therefore, this study can assume that the regression model for differences is effective during daytime hours (6:00–18:00). With the advantage of the temporal continuity of the ERA5-Land data, the 1 km hourly air temperature data can be obtained by subtracting the predicted differences from the ERA5-Land data (regression model B).

In the whole downscaling model, the GP algorithm is employed to effectively merge data from multiple sources by constructing a statistical regression model for downscaling. For further elaboration, refer to Section 3.2. Notably, the air temperature data recorded by the weather station serve as both the target variable and the verification data for evaluating

the performance of the regression model in this study. See Section 3.3 for the data allocation of the training set and the validation set.

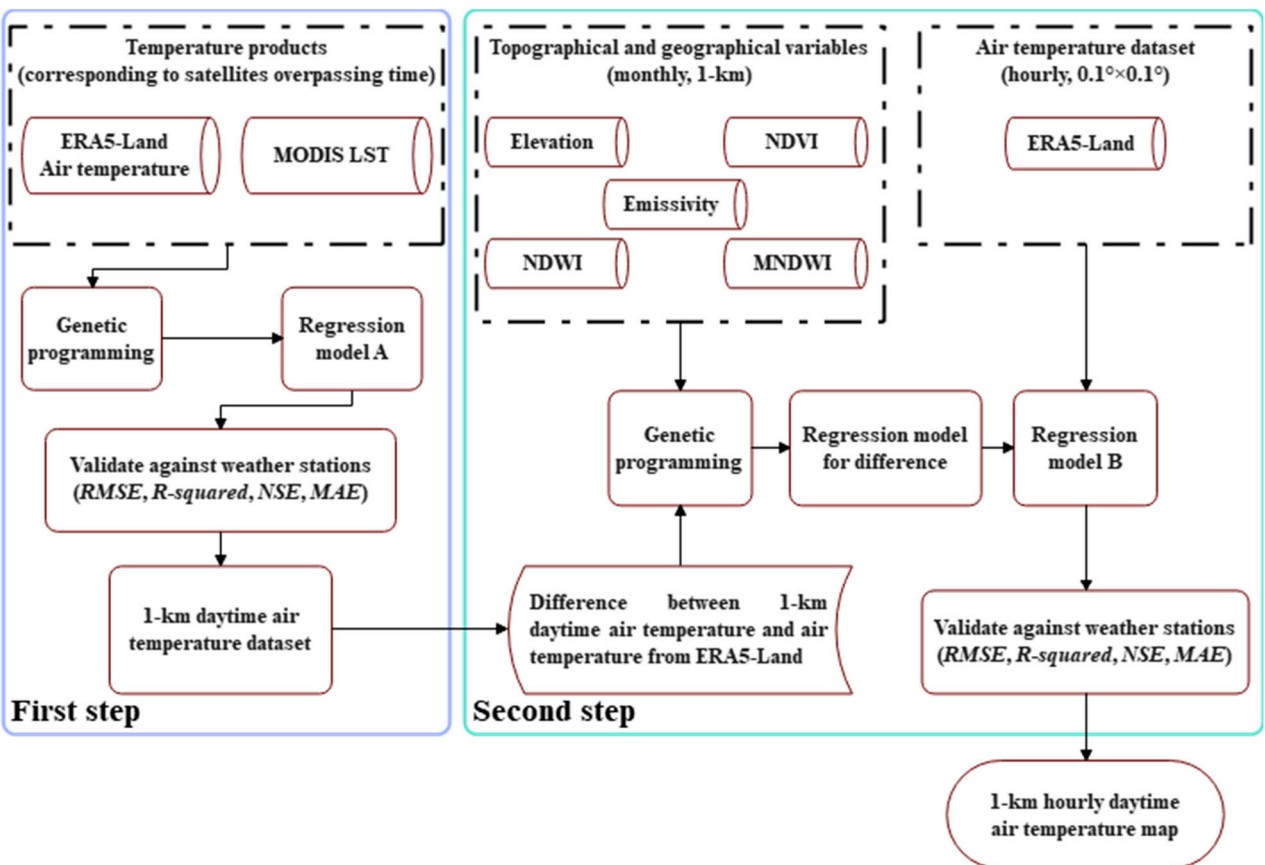

**Figure 2.** The flowchart of the statistical downscaling model generation process.

### 3.2. Genetic Programming (GP) Algorithm

In this research, a GP toolbox in MATLAB was employed to fuse multi-source input data from and generate statistical regression models for downscaling purposes [45]. The schematic diagram of the GP algorithm is presented in Figure 3. Since the algorithm would generate many potential solutions during the process of running the binary tree structure, a fitness function has been used to evaluate the complexity and accuracy of obtained regression models, as shown in Equation (1):

$$f = \frac{r}{1 + \exp[a_1(L - a_2)]} \tag{1}$$

where $f$ is the calculated fitness value, $r$ is the correlation coefficient, and $L$ is the number of nodes in the binary tree. $a_1$ and $a_2$ are penalty coefficients utilised to decrease the fitness values of binary trees with complex terms. The key parameters of the GP algorithm for this study are shown in Table 3. More details about the algorithms and the toolbox can be found in the report of Madár et al. [46].

**Table 3.** The key parameters of the GP algorithm.

| Initial Population Size | Maximum Tree Depth | Crossover Probability | Mutation Probability | Selection Type | Iterations Number |
|---|---|---|---|---|---|
| 500 | 15 | 0.7 | 0.3 | Tournament selection | 30 |

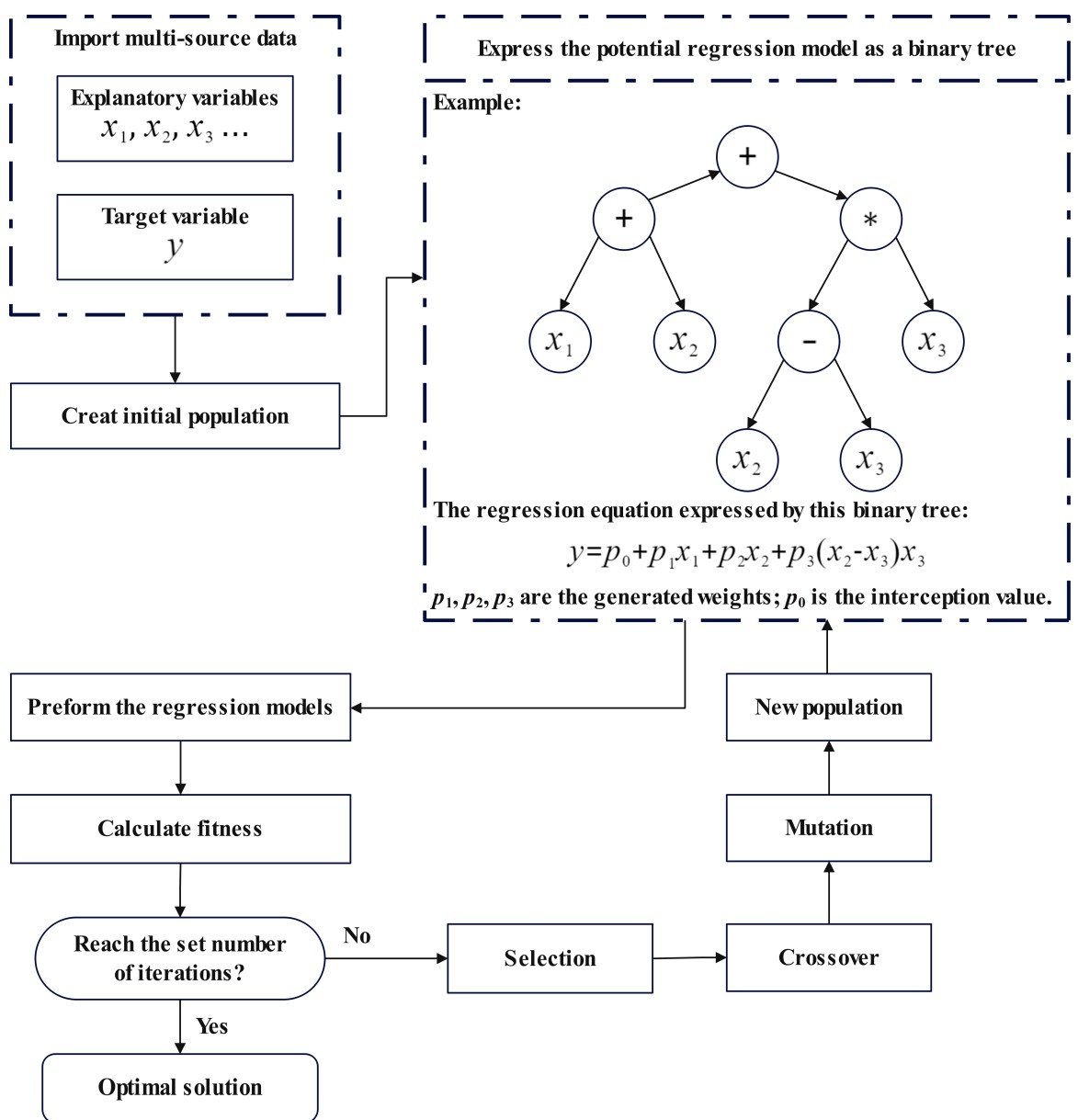

**Figure 3.** The schematic diagram of the GP toolbox.

### 3.3. Model Fitting and Statistical Indicators

To ensure the accuracy of the generated air temperature data, the hold-out method was adopted to train statistical regression models for both model A and B. Two-thirds of the data were randomly selected as the training set to train the model, and one-third of the data as the validation set to evaluate the performance of the model [47]. The method can also check whether the trained regression model has an overfitting phenomenon simultaneously. Moreover, the forward stepwise regression (FS) method was utilised to rank the significance of these explanatory variables. It starts from an empty model, adding one explanatory variable that fits the model at a time until some stopping criterion is satisfied [48]. In this case, using all explanatory variables was considered as the stopping criterion of the model.

To evaluate the regression model, four statistical indicators commonly used in atmospheric science were adopted, including the mean absolute error (*MAE*), the *RMSE*, the *R-squared*, and the Nash–Sutcliffe efficiency coefficient (*NSE*). The equations are as follows:

$$MAE = \frac{1}{T}\sum\nolimits_{t=1}^{T}\left|Q_o^t - Q_s^t\right| \tag{2}$$

$$RMSE = \sqrt{\frac{1}{T}\sum_{t=1}^{T}(Q_o^t - Q_s^t)^2} \tag{3}$$

$$R-squared = \left(\frac{cov(o,s)}{\sigma_o\sigma_s}\right)^2 \tag{4}$$

$$NSE = 1 - \frac{\sum_{t=1}^{T}(Q_o^t - Q_s^t)^2}{\sum_{t=1}^{T}(Q_o^t - Q_o)^2} \tag{5}$$

where $Q_o^t$ and $Q_s^t$ represent observed air temperature data and simulated air temperature data at time $t$, respectively. $cov(o,s)$ is the covariance between observed air temperature and estimated air temperature data. $\sigma_o$ and $\sigma_m$ are the standard deviation of observations and the regression model, respectively.

## 4. Results and Discussions

### 4.1. The Performance of Regression Models

As shown in Figure 2, the first step of the proposed downscaling model fused the air temperature data from ERA5-Land and the LST data from MODIS. This process aimed to generate an optimal statistical regression model (regression model A) for obtaining air temperature data in 1 km cloud-free pixels in the study area corresponding to the satellite overpassing time. The obtained optimal regression model is found as follows:

$$y = 0.934x_1 - 0.057x_2 + 0.003x_2^2 + 1.437 \tag{6}$$

where $y$ is the predicted value of air temperature, $x_1$ is the air temperature from ERA5-Land, and $x_2$ is the LST from MODIS satellites. In this study, the performance of our obtained regression models was evaluated using station-based data as a benchmark. The evaluation results can be seen in Table 4. It demonstrated that the air temperature predicted by regression model A exhibited a strong correlation with in situ ground observations, with an *R-squared* value of 0.931, an *RMSE* of 0.070 °C, an *MAE* of 0.884 °C and an *NSE* of 0.930.

**Table 4.** The performance of regression model A.

| Stations | *R-squared* | *RMSE* (°C) | *MAE* (°C) | *NSE* |
|---|---|---|---|---|
| Hampstead | 0.941 | 0.158 | 0.616 | 0.939 |
| Heathrow | 0.925 | 0.288 | 0.957 | 0.920 |
| Kenley Airfield | 0.954 | 1.110 | 1.175 | 0.892 |
| Kew Gardens | 0.950 | 0.015 | 0.706 | 0.950 |
| London St James's Park | 0.953 | 0.300 | 0.739 | 0.946 |
| Northolt | 0.933 | 0.140 | 0.841 | 0.932 |
| Full validation set (all stations) | 0.931 | 0.070 | 0.884 | 0.930 |

The histogram of the residuals created based on the regression model A is presented in Figure 4a, which displays a normal distribution behaviour with a classic bell shape. The normal distribution behaviour indicates that the regression model confirms the distribution law of continuous data in nature, which means the regression model is reasonable enough. The temporal variations of air temperature data from weather stations and regression model A are presented in Figure 5, demonstrating that regression model A is in good agreement with the station-based observations. Moreover, Figure 5 also indicates that regression model A can accurately estimate air temperature during extremely hot periods in summer. Thus, the presented results demonstrated the strong consistency and accuracy between regression model A and the meteorological station data, which can represent the actual conditions effectively.

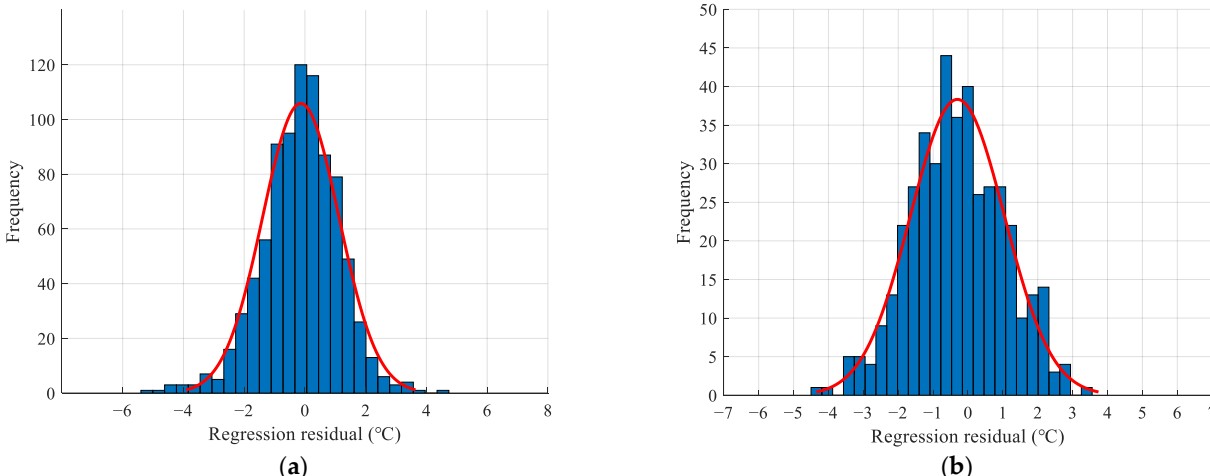

**Figure 4.** Model residuals: (**a**) regression model A; (**b**) regression model B.

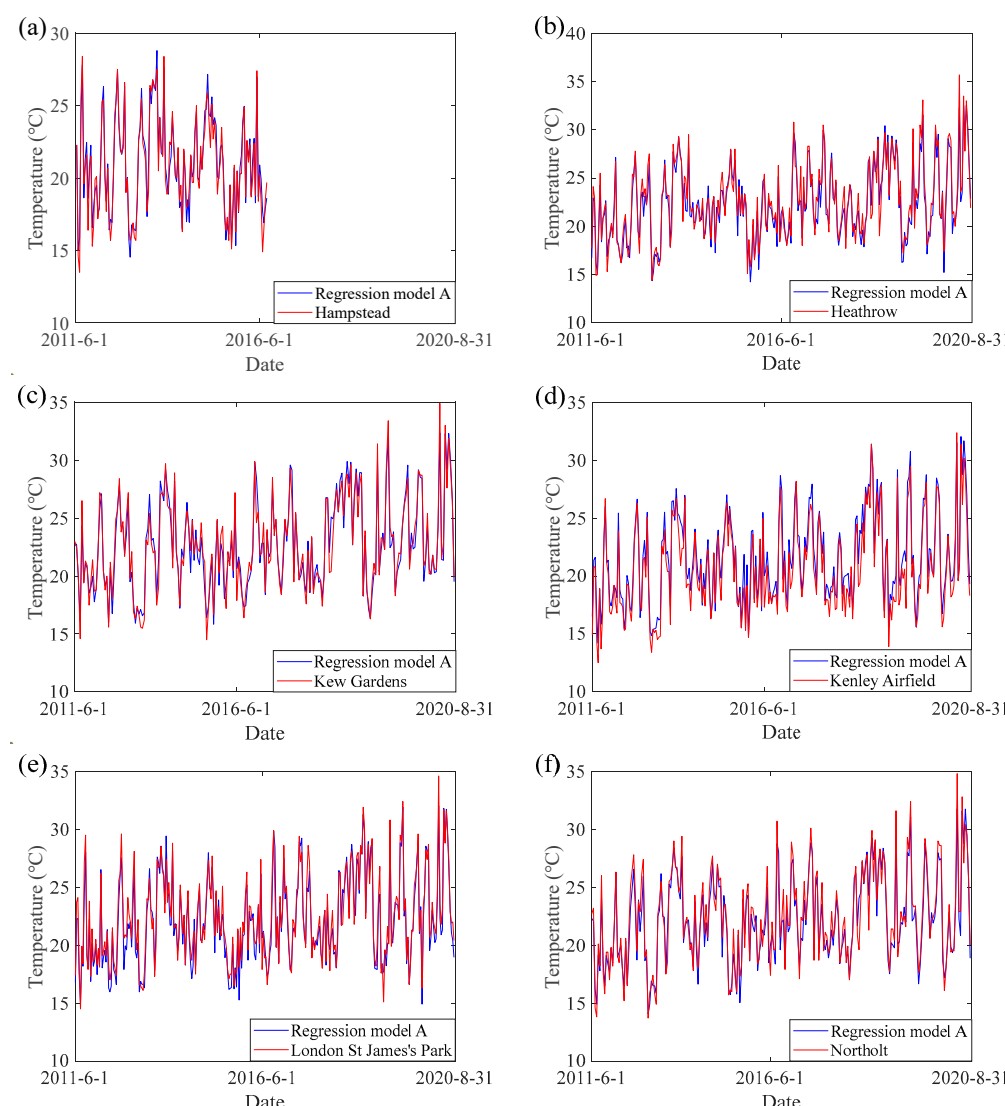

**Figure 5.** Temporal variations of hourly air temperature during daytime (6:00–18:00) in the regression model A and weather stations: (**a**) Hampstead; (**b**) Heathrow; (**c**) Kew Gardens; (**d**) Kenley Airfield; (**e**) London St James's Park; (**f**) Northolt (please note that these figures only cover available time points from 2011 to 2020).

Due to the extremely high accuracy of the air temperature obtained from regression model A, it was considered in this study to represent the actual air temperature of each pixel in the study area. In the second step of the downscaling model, five topographical and geographic variables were utilised to construct a statistical regression model (regression model B) that can explain the differences between the actual air temperature data and the air temperature data from ERA5-Land. Considering the stable weather conditions during the heatwaves, the researchers assumed that regression model B was valid during daytime (6:00–18:00). The optimal regression model B obtained by the GP algorithm can be seen as follows:

$$
\begin{aligned}
T = t - 3.005p_1 \quad &-0.273p_2 + 0.831p_3 - 48.876p_4 + 374.662p_1^2 - 1.768p_3^2 + 193.127p_1p_2 \\
&+0.001p_4(p_4 - p_5) - 194.023p_1p_2p_4 - 378.215p_1^2p_4 - 1.397p_1^2p_2^2p_3 \\
&+\frac{0.278p_1^2}{p_2p_4(p_4-p_1)} + 49.520
\end{aligned}
\tag{7}
$$

where $T$ is the predicted value of 1 km hourly air temperature, $t$ is the hourly air temperature data from ERA5-Land, $p_1$ is the MNDWI from Landsat 8 products, $p_2$ is the NDVI from MODIS products, $p_3$ is the NDWI from Landsat 8 products, $p_4$ is the emissivity from MODIS products, and $p_5$ is the elevation from STRM products. The performance of regression model B as shown in Table 5, which was also validated using in situ ground observations as a benchmark. The results demonstrate a high level of accuracy, with the *R-squared* value of 0.949, *RMSE* of 0.335 °C, *MAE* of 1.115 °C, and *NSE* of 0.924.

**Table 5.** The performance of regression model B.

| Stations | R-squared | RMSE (°C) | MAE (°C) | NSE |
|---|---|---|---|---|
| Hampstead | 0.931 | 0.821 | 1.203 | 0.889 |
| Heathrow | 0.973 | 0.113 | 1.015 | 0.939 |
| Kenley Airfield | 0.917 | 1.055 | 1.374 | 0.856 |
| Kew Gardens | 0.965 | 0.518 | 1.108 | 0.925 |
| London St James's Park | 0.978 | 0.463 | 0.821 | 0.955 |
| Northolt | 0.969 | 0.400 | 1.279 | 0.916 |
| Full validation set (all stations) | 0.949 | 0.335 | 1.115 | 0.924 |

Figure 4b shows the histogram of the residuals created based on regression model B. It also displayed a normal distribution behaviour with a classic bell shape. To better display the temporal variations between station-based air temperature data and estimated air temperature data from regression model B, a relatively long cloud-free period is required. Therefore, the period from 23 August to 29 August 2019, was selected to display the temporal variations and spatial distribution patterns. The results of temporal variations depicted in Figure 6 revealed a strong agreement between the estimated air temperature data from regression model B and the station-based observations. Figure 7b presents the spatial distribution of air temperature at 12:00 on 23 August 2019 (cloud-free example), which is generally consistent with the spatial distribution of estimated air temperature data from regression model A in Figure 7a.

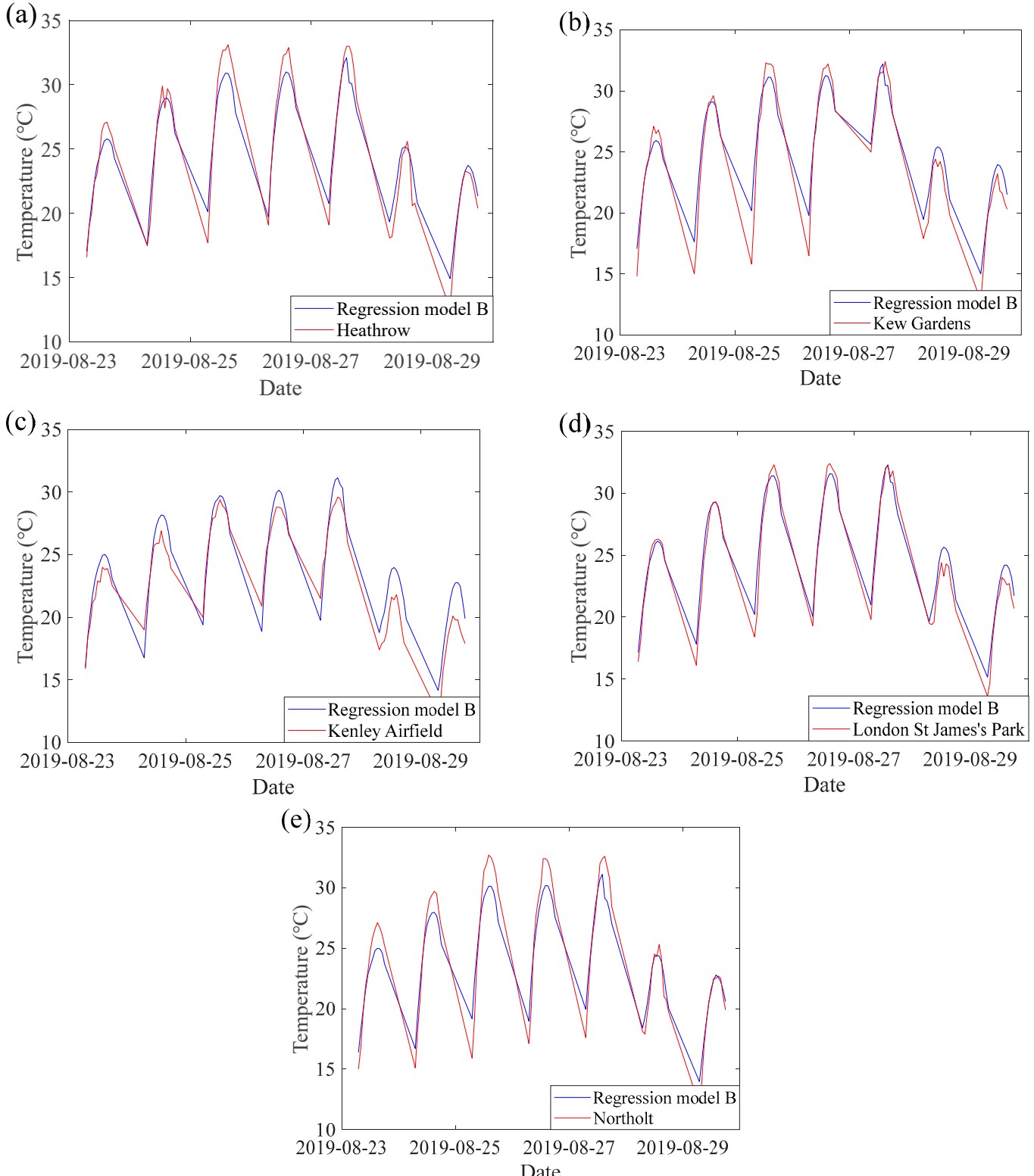

**Figure 6.** Temporal variations of hourly air temperature during daytime (6:00–18:00) in regression model B and weather stations: (**a**) Heathrow; (**b**) Kew Gardens; (**c**) Kenley Airfield; (**d**) London St James's Park; (**e**) Northolt.

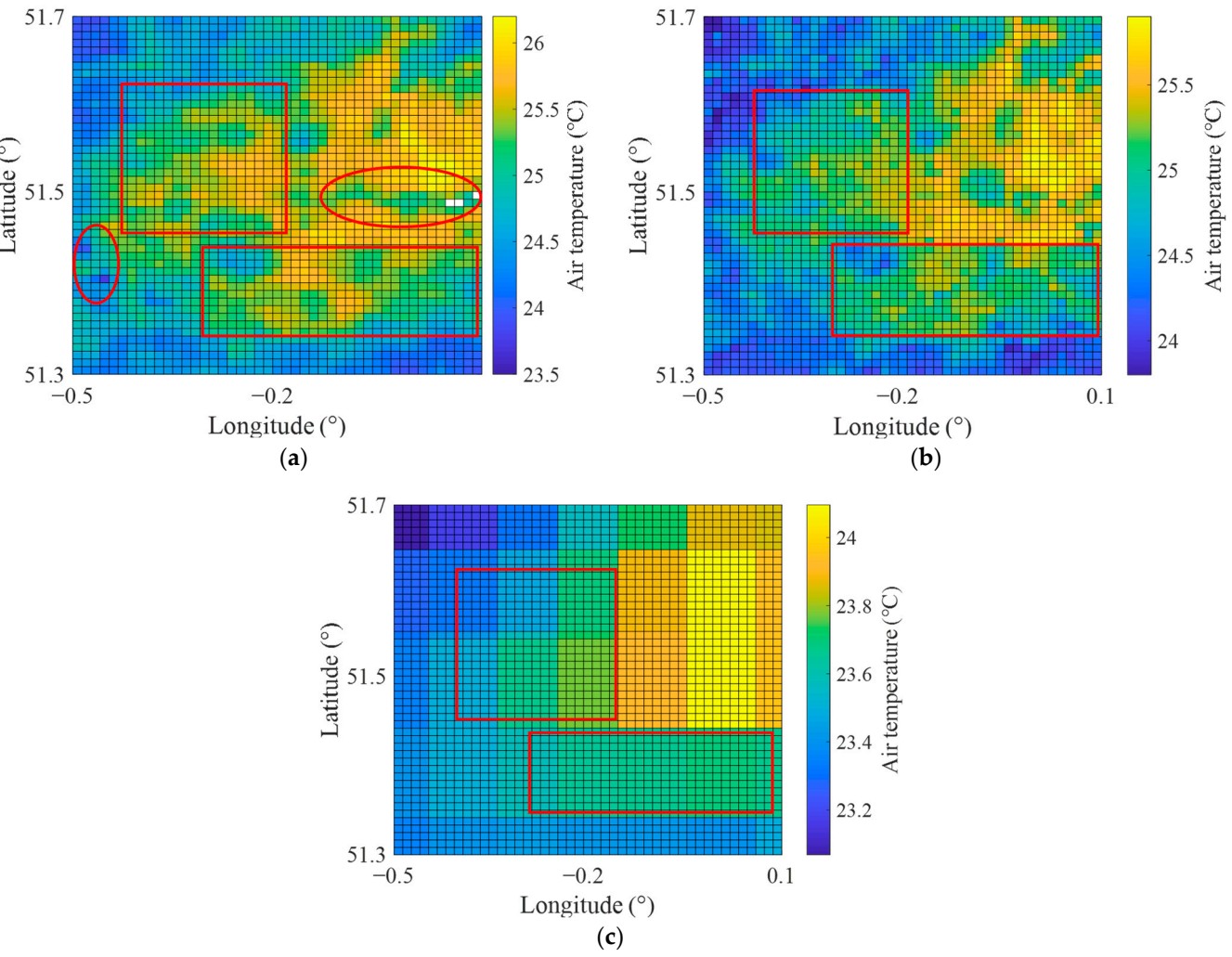

**Figure 7.** The spatial distribution patterns across London on 23 August 2019 at 12:00: (**a**) estimated air temperature data from regression model A; (**b**) estimated air temperature from regression model B; (**c**) air temperature data from ERA5-Land.

Dos Santos also focused on estimating summer high temperatures in the London urban area. Although Dos Santos adopted machine learning methods to obtain daily maximum air temperature, the performance of the regression model remained less reliable, with the *RMSE* of 2.03 °C and *R-squared* of 0.68 [15]. On the other hand, due to its utilization of daily satellite data (e.g., LST, Black Sky Albedo (BSA)) as explanatory variables for estimating daily air temperature, this method is unable to generate hourly air temperature data. The studies of Li et al. and Janatian et al. reported similar findings. Li et al. developed geographically weighted regression (GWR) models, which used 1 km daily LST and elevation as explanatory variables to estimate daily air temperature [40]. Janatian et al. adopted 13 explanatory variables from MODIS products to explore a statistical model for estimating air temperature based on multiple linear regression [49]. Although both methods performed well (*R-squared* > 0.90), they could not estimate hourly temperature due to the temporal resolution of explanatory variables (daily). In comparison, our proposed approach can effectively compensate for their shortcomings in terms of accuracy and temporal continuity. Furthermore, the comparison with the studies conducted by Zhou, B. et al. also demonstrated positive results. Similar to our method, Zhou, B. et al. utilized a two-stage machine learning approach to develop statistical downscaling models for estimating 1-km air temperature [50]. The results, with an *RMSE* of 1.58 °C and *R-squared* of 0.96, demonstrated the significant potential of their method in estimating high spatiotemporal

resolution air temperature. However, their proposed two-stage framework, which only focused on estimating air temperature at satellite overpassing time, was still insufficient for observing temperature variations within a day. Our proposed novel two-stage method filled in these weaknesses and allowed for further exploration of estimating continuous, hourly air temperature while maintaining high accuracy. Zhou, S. et al. explored the use of explanatory variables to build an interpretable hourly air temperature estimation model based on light gradient boosting machine (LightGBM) [51]. However, since this model relies on data from geostationary meteorological satellites, it cannot be applied globally. Using satellite products that provide global data may potentially become a future research direction for this method. Chen, G. et al. and Chen, S. et al. proposed the use of random forest models to obtain high-resolution hourly air temperature, demonstrating good performance with *R-squared* of 0.80 and 0.96, respectively. However, this method heavily relied on high-density meteorological station data. In the research of Chen, G. et al., they utilized observed data from 86 meteorological stations for Kriging interpolation to estimate spatial patterns for driving the random forest model [52]. Chen, S. et al. collected data from meteorological stations over a period of three days, totalling 218 stations, and used the random forest model to generate regression models for estimating the air temperature of each pixel [53]. Carrión et al. reported similar limitations. Based on the XGBoost machine-learning algorithm, their team developed a statistical model to estimate hourly air temperature using explanatory variables such as LST and EVI [54]. To conduct spatial cross-validation, their study collected data from around 4000 meteorological stations. Clearly, these approaches are not suitable for cities with a sparse distribution of meteorological stations, such as London, and our proposed model can effectively overcome this limitation.

### 4.2. Inadequacies of ERA5-Land Data during Downscaling

Figure 7c presents the spatial distribution pattern of air temperature obtained from ERA5-Land at the time corresponding to Figure 7b. Compared with Figure 7a, it can be found that ERA5-Land data presented anomalously low-temperature patterns in many built-up areas, such as the marked red rectangular areas. Combining with the patterns of Figure 7b,c, the abnormally low-temperature regions were caused by the distribution characteristics of ERA5-Land data. Specifically, urban areas of London only had 35 ERA5-Land pixels at 0.1° resolution in this study. Although ERA5-Land observations have been shown to have high accuracy at the country level in previous studies [55], the complex urban surface can result in considerable spatial temperature differences at the microenvironment scale. Furthermore, compared to forests and croplands, ERA5-Land in built-up areas had the lowest accuracy in air temperature estimations [11]. Although ERA5-Land data showed promising performance in air temperature estimation in this case, as evidenced by the convincing results in Table 6 (*R-squared* = 0.945, *RMSE* = 0.953 °C, *MAE* = 1.355 °C and *NSE* = 0.888), its spatial distribution patterns cannot adequately represent the built-up lands at the city level. As a result, using ERA5-Land data alone as a global-scale variable in the downscaling model would inevitably result in abnormal low-temperature areas. Therefore, considering other variables was necessary for this downscaling modelling.

**Table 6.** Performance results of models during the forward stepwise regression.

| No. | Description | Items | Statistical Indicators | | | |
|-----|-------------|-------|-----------|------|------|------|
| | | | *R-squared* | *RMSE* | *MAE* | *NSE* |
| 1 | Null variable | (ERA5-Land) | 0.945 | 0.953 | 1.355 | 0.888 |
| 2 | | Elevation | 0.953 | 0.506 | 1.129 | 0.922 |
| 3 | | Emissivity | 0.943 | 0.528 | 1.194 | 0.912 |
| 4 | One variable | NDVI | 0.951 | 0.481 | 1.139 | 0.921 |
| 5 | | NDWI | 0.946 | 0.499 | 1.178 | 0.915 |
| 6 | | MNDWI | 0.943 | 0.459 | 1.179 | 0.915 |

**Table 6.** *Cont.*

| No. | Description | Items | Statistical Indicators | | | |
|-----|------------|-------|-----------|------|-----|-----|
| | | | *R-squared* | *RMSE* | *MAE* | *NSE* |
| 7 | Two variables | Elevation + Emissivity | 0.947 | 0.530 | 1.171 | 0.916 |
| 8 | | Elevation + NDVI | 0.953 | 0.478 | 1.125 | 0.924 |
| 9 | | Elevation + NDWI | 0.951 | 0.486 | 1.139 | 0.921 |
| 10 | | Elevation + MNDWI | 0.952 | 0.472 | 1.128 | 0.924 |
| 11 | Three variables | Elevation + NDVI + Emissivity | 0.949 | 0.521 | 1.156 | 0.919 |
| 12 | | Elevation + NDVI + NDWI | 0.952 | 0.353 | 1.094 | 0.927 |
| 13 | | Elevation + NDVI + MNDWI | 0.954 | 0.344 | 1.080 | 0.929 |
| 14 | Four variables | Elevation + NDVI + MNDWI + Emissivity | 0.948 | 0.392 | 1.129 | 0.922 |
| 15 | | Elevation + NDVI + MNDWI + NDWI | 0.954 | 0.329 | 1.083 | 0.929 |
| 16 | All variables | Elevation + NDVI + MNDWI + NDWI + Emissivity | 0.949 | 0.335 | 1.115 | 0.924 |

### 4.3. Challenges Posed by Water Bodies

Figure 8 displays the spatial distribution pattern of topographical and geographic variables in August 2019. As illustrated, the emissivity data can well distinguish built-up areas from water body areas and forested areas at 1 km resolution. For the images of NDWI and MNDWI, water body areas are marked in bright yellow. Given that surface composition can significantly affect the LST of the study area, NDWI and MNDWI have been frequently used in previous temperature-related studies to describe the distribution of water bodies [56]. However, upon comparing the areas in Figure 7a corresponding to the red oval areas in Figure 8d,e, we discovered that the low air temperature area caused by water bodies is much larger than what is presented by the NDWI and MNDWI images. Hathway and Sharples [57] had similar findings, indicating that water body areas such as rivers can effectively cool their surrounding environment. Furthermore, it should be noted that although Figure 8d,e emphasised the distribution of water bodies, the values for the water body areas and the built-up areas were still relatively close in the image. As an example, in NDWI images, the values for water body areas ranged from −0.2 to 0, while those for built-up areas ranged from 0 to 0.2. Comparable results were also observed in MNDWI images. Due to these similarities, distinguishing water body areas from built-up areas in downscaling models became challenging. Consequently, to better characterise the cooling effect of water bodies on their surrounding environment and distinguish different land covers, it was necessary to introduce other variables, such as emissivity, to compensate for this limitation.

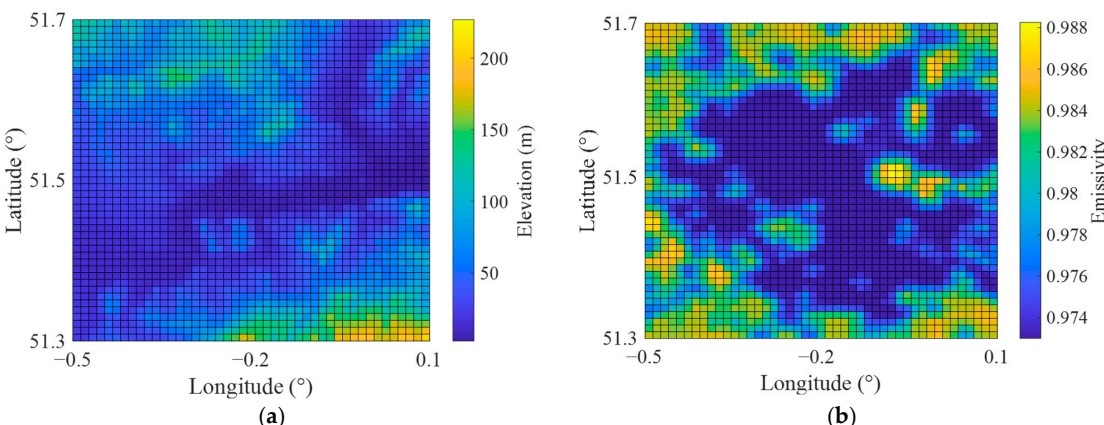

**Figure 8.** *Cont.*

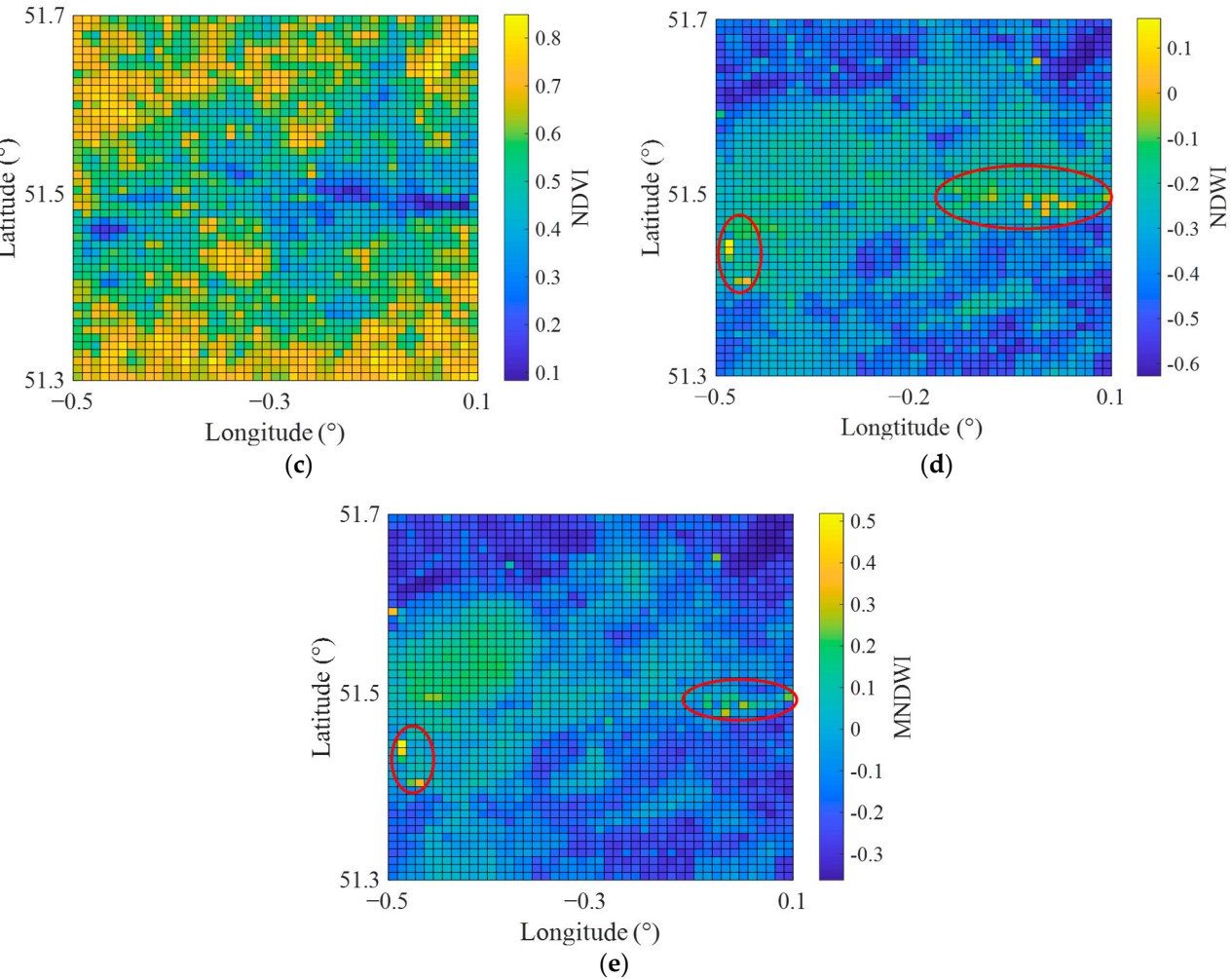

**Figure 8.** The spatial distribution patterns of topographical and geographical variables across London in August 2019: (**a**) elevation from SRTM; (**b**) emissivity from MODIS; (**c**) NDVI from MODIS; (**d**) NDWI from Landsat 8; (**e**) MNDWI from Landsat 8.

### 4.4. The Results of Forward Stepwise Regression

Table 6 shows the forward stepwise regression results, providing performance metrics (*R-squared*, *RMSE*, *MAE* and *NSE*) of 16 regression models for comparison. As shown in Table 6, the elevation significantly impacted daytime air temperatures, with an *NSE* value of 0.922. Even if we subsequently added other variables (i.e., elevation, NDVI, NDWI, MNDWI), the *NSE* value of the models did not exceed 0.929. Similar findings were also reported by Abunnasr and Mhawej [8]. As the explanatory variables describing the spatial distribution pattern of water bodies, NDVI and MNDWI were regarded as the second and third most important variables, respectively. During the forward stepwise regression modelling, emissivity was the final variable selected and was deemed the least influential among all the explanatory variables. Furthermore, even though the four-variable models (comprising elevation, NDVI, NDWI, and MNDWI) exhibited nearly identical performance to the all-variable models, the latter was still adopted as the regression model B. This decision was made based on the understanding that the all-variable models provide a more comprehensive explanation of local-scale climate characteristics.

It was slightly unexpected to see that, although the spatial distribution pattern of emissivity was most consistent with that of LST compared to other explanatory variables, it was classified as the least influential variable in Table 6. It is mainly attributable to the slight difference in emissivity between different land cover types. For instance, the values for built-up areas ranged from 0.972 to 0.974, while those for both water body areas and

forested areas ranged from 0.982 to 0.988. Thus, although the built-up area can be well differentiated, water body areas and forested areas were indistinguishable in emissivity images due to numerical similarities. Similar issues were also reported by Zou et al. [11], who found that emissivity can distinguish built-up areas from natural areas (i.e., grasslands, forests, and water body areas) but with minimal differences. In addition, the limited number of weather stations in urban areas made emissivity data poorly calibrated. The existing six weather stations were far from meeting the requirements for verifying the 10 land covers shown in Figure 1b, which meant the data lacked representativeness. Therefore, it made monthly emissivity less valuable than other variables in this case. Moreover, it also should be noted that surface emissivity is sensitive to precipitation [38]. Although previous studies proved that emissivity could be assumed to remain unchanged over a period of time [58], precipitation (i.e., dew, rain, and snow) can significantly affect emissivity values over a few days. Hence, incorporating precipitation-related variables such as soil moisture and rainfall as explanatory variables for air temperature downscaling may further improve estimation performance.

For error metrics, as can be seen from Table 6, among the four types of statistical indicators, *R-squared* and *NSE* have a slight variation range for different regression models, which varies between 0.945 and 0.954 and between 0.888 and 0.929, respectively. Although the range of *RMSE* and *MAE* was larger compared with *R-squared* and *NSE*, the difference between the one-variable and all-variables models was still very small. The change in *RMSE* was the most obvious, as it had a clear downward trend from the one-variable model to the all-variables model. It is worth noting that many atmospheric science studies prefer to use *R-squared* as a metric for error performance [11,21]. However, our results observed that the variation range in *R-squared* does not match the changes in spatial distribution patterns during the forward stepwise regression process. For example, Figure 7a,b indicate the spatial distribution patterns of air temperature from regression model B and ERA5-Land, respectively. However, there was no significant difference in their *R-squared* value, as shown in Table 6 (No. 16 and No. 1, respectively). This is mainly because both dependent and independent variables tended to change over time, which can lead to inflated *R-squared* values. Due to this, the performance of ERA5-Land data in terms of *R-squared* values was too good to make it challenging to observe the improvements brought about by adding other explanatory variables to the downscaling model. Moreover, while the model results exhibit high *R-squared* values, it is worth noting that such values do not always indicate a good model [59], as they cannot measure predictive error. This observation was also reported by Jia et al. [60], who found that the best model fitting cannot always result in the best downscaling outcomes. Therefore, relying on *R-squared* alone as an indicator for the correlation of the proposed downscaling method is insufficient, and more attention should be given to other validation indicators (e.g., *RMSE*).

*4.5. Limitations of the Research*

It cannot be ignored that the research still has some limitations. As seen in Figure 6, while comparing the temporal variations of hourly air temperature during daytime (6:00–18:00) in regression model B and weather stations, large differences occurred at the maximum and minimum values. This is mainly based on the limitations of the following two aspects. First, regression models cannot completely correctly estimate temperature data in extreme cases, and the large difference is prone to occur when estimating maximum or minimum values. Many temperature-related statistical downscaling studies have also reported similar problems [8,22]. Second, many studies have included the Sun Zenith Angle as one of the explanatory variables, because it can explain the ability of land surface to absorb solar energy at different times. Therefore, the assumption made in this study that regression model B is always effective during daytime hours (6:00–18:00) may introduce some errors. Furthermore, upon examining Table 5 and Figure 6, notable differences can be observed in the errors and temporal variations of Kenley Airfield compared to other meteorological stations. This discrepancy can primarily be attributed to the fact that, in the first step

of the downscaling model depicted in Figure 3, it focused solely on investigating the potential functional relationship between LST and air temperature using data from the available six meteorological stations, with Kenley Airfield being the sole station situated in the urban outskirts as illustrated in Figure 1. Considering that urban outskirts often exhibit distinctive geographical and environmental conditions, and the influence of human activities and urbanization is reduced, the statistical regression model employed to estimate air temperature in these areas may introduce additional errors. The significant errors observed in Kenley Airfield, as indicated in Table 4, further confirm this point. Moreover, as shown in Figure 1, the limited number of stations cannot fully represent the environmental conditions of all land covers. This is a major weakness of statistical downscaling approaches because it is based on the representativeness of meteorological station data. Although this study explored the possibility of quantifying the impact of land cover on air temperature by introducing emissivity, the existing stations may still be insufficient to generate convincing regression models. Consequently, future research should aim to expand the study area and gather more comprehensive weather station data, potentially enhancing the performance of the existing statistical regression model.

Cloud cover is also an important factor that can easily introduce errors, especially for Landsat satellites that take 16 days to complete a scan of the globe [12]. Therefore, the data of NDWI and MNDWI from Landsat products may lead to data gaps due to the effect of cloud cover. In future research, we plan to explore more suitable satellite products (such as the CGLS-LC100 Collection 3) and radar data (such as the 1 km Resolution UK Composite Rainfall Data from the Met Office Nimrod System) to replace the NDWI data from Landsat and analyse the influence of water bodies and precipitation on air temperature.

## 5. Conclusions

In this research, we proposed a new two-step data fusion model to produce temporally continuous, high spatial-temporal resolution air temperature data. Using London as a case study, the hourly air temperature at 1 km resolution during daytime (6:00–18:00) was successfully obtained by fusing satellite and reanalysis datasets with station-based observations. The two-step downscaling model based on the GP algorithm demonstrated superior performance in obtaining air temperature data in London as compared to other similar studies. It achieved good performance with the *RMSE* of 0.335 °C, *R-squared* of 0.949, *MAE* of 1.115 °C, and *NSE* of 0.924, surpassing previous studies and demonstrating its potential in estimating hourly air temperature data. Compared to other downscaling models that can only obtain daily temperature data, the proposed model can provide better temporal continuity while maintaining high accuracy, allowing for estimating hourly air temperature data during heatwave events.

The significance of explanatory variables was ranked using the forward stepwise regression model. The results showed that elevation considerably impacted the spatial distribution of air temperature, while emissivity was the least influential variable. This was primarily because emissivity values were numerically similar across different land covers, making it difficult to distinguish some land covers (e.g., forested areas and water body areas) in emissivity images. Additionally, the sensitivity of surface emissivity to precipitation was another factor that could affect the values. Thus, adding precipitation-related variables such as soil moisture and rainfall as explanatory variables may provide a potential improvement solution. The performance of four error metrics revealed the limitation of *R-squared* in the downscaling model, which is the limited variation range and inflated *R-squared* values. Therefore, in future research, more attention should be given to other validation indicators, such as *RMSE*. Furthermore, using ERA5-Land data as a global-scale variable for downscaling in urban areas can inevitably result in spatial differences in air temperature at the microenvironment scale due to the complex surface of urban areas. Although there is a slight disadvantage, our results demonstrated that the proposed multi-source data fusion model could generate high-quality air temperature data suitable for heatwave-related studies. Given the limited number of meteorological stations in urban

areas, the produced air temperature datasets have important implications for public health research, which requires quantitative data, especially continuous and high-resolution data, to support excess mortality analysis associated with heatwaves. Moreover, the resulting dataset can also provide valuable support for researching the environmental impacts of urbanisation, such as the UHI effect and its implications on building energy consumption and human health.

**Author Contributions:** Methodology, Z.W. and L.Z.; Software, Z.W., Q.W., J.W., Y.L., S.D., A.A. and D.H.; Validation, Z.W.; Resources, Z.W.; Data curation, Q.W.; Writing—original draft, Z.W.; Writing—review & editing, Z.W., L.Z. and D.H.; Supervision, L.Z. and D.H. All authors have read and agreed to the published version of the manuscript.

**Funding:** This study is supported by Resilient Economy and Society by Integrated SysTems modeling (RESIST), Newton Fund via Natural Environment Research Council (NERC) and Economic and Social Research Council (ESRC) (NE/N012143/1).

**Data Availability Statement:** The data and the code of this study are available from the corresponding author upon request.

**Conflicts of Interest:** The authors declare no conflict of interest.

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
