# Peer review of "Data Fusion for Estimating High-Resolution Urban Heatwave Air Temperature"

_remotesensing, doi:10.3390/rs15163921_

Round 1

Reviewer 1 Report

The submitted manuscript has both good language and flow (with minor grammatical issues). However, there are several concerns, mainly regarding the novelty, data preparation, and experimental design. These concerns are very common to all papers in this category: we take samples from ground station data & Earth remote sensing data, apply a well-known algorithm (e.g., found in MATLAB, without modifications), and feel that the obtained results are new and significant.

At the beginning, the manuscript is reassuring, and the reader can hope for a breakthrough as Authors write “the limited number of meteorological stations restricts their ability to present the spatial distribution of air temperature, especially in urban areas” which is, of course, true. However, starting from the section that describes the data sampling, the reader begins to realize that nothing substantially new is going to be presented.

DATA SAMPLING/PREPARATION, EXPERIMENTAL DESIGN

The idea of combining satellite and in-situ data is not new, but important. However, the manuscript follows a very straightforward scheme for the data fusion: it just takes measurements of several ground meteorological stations and cell/pixel values of MODIS, ERA5, and other datasets that cover these stations.

The number of pixels/cells of MODIS (even with 1km of spatial resolution) that cover the area of interest (London) incredibly exceeds the number of measurements of this tiny set of meteorological stations.

Hence, it is not surprising that (like in other similar papers):

a) Authors obtain a catastrophically small training as well as validation sample compared to the overall available satellite data

b) it is simply impossible to validate the performance of the obtained model (regression in this case) on other cell/pixel values of MODIS outside the locations of these stations using the presented technique

Authors write “the data were collected from 1-km pixels corresponding to six meteorological stations, resulting in a total of 2764 samples.” There are thousands of pixels/cells of MODIS rasters that cover the area of interest (London); the Authors do not state explicitly how many pixels/cells are there (it is obvious that the number of such pixels is much larger compared to ground stations). Among those, only 2764 of values were obtained.

Hence, as the data sampling technique is not very sophisticated, it is very easy to reason about what one can expect in the remainder of the manuscript – good results only for locations of these ground stations.

A great natural question arises: why do we need the presented technique and this research at all? To predict values at locations of ground stations using MODIS, and other datasets? But we can do better with numerical weather prediction or other techniques. To predict meteorological values at 2m above the ground using MODIS, and other datasets, but for cells/pixels that do not cover these stations? But it is impossible to validate the performance of the proposed model on other cells/pixels using the presented technique.

MORE ON NOVELTY, SIGNIFICANCE & THE ALGORITHM

Authors write “In this research, a GP toolbox in MATLAB was employed to fuse multi-source input data from and generate statistical regression models for downscaling purposes [39].” Well, this is another evidence that the manuscript does not bring much novelty. Why this research is significant? It performs a super-simple sampling, runs a MATLAB algorithm (both a student can do), and reports reliable results only for a tiny number of locations for which we already know the values (an obvious result).

SUMMARY

For a flagship journal, there are: weak novelty, limited experiments, scope and results, and unclear contribution to state-of-the-art.

Minor edits are required

Reviewer 2 Report

The authors present a two-step downscaling model to generate hourly air temperature data at 1km resolution for urban heatwave research. The modeling approach is sound and the validation method is effective. However, the question is that the ERA5-Land data with 0.1° resolution and the MODIS data with 1km resolution are not the best data for the inner urban study. Their resolution is too coarse to reflect the spatial heterogeneity within the city as the results show in Fig.7. Many studies have combined Landsat and MODIS to develop spatiotemporal downscaling algorithms to obtain higher spatiotemporal resolution products. Is it possible to adopt the two-step model proposed in this study to obtain air temperature data with higher spatiotemporal accuracy? The significance of this study is obvious, but the results seem to be unable to fully achieve the research objectives. Further experiments are needed. The specific suggestions are as follows:

1.      Line 25, the authors concluded that compared with other downscaling methods, the two-step model proposed in this study achieved better accuracy. However, this is a conclusion drawn solely from the error indicators. Using the same dataset and adding comparison results evaluated by other downscaling algorithms would be more convincing for this conclusion.

2.      Line 115, There is extra paragraph indentation here.

3.      Line 145, In Fig.1(b), the data source of the land cover should be illustrated.

4.      In section 2.2, the 16-day NDVI data from MODIS was used for model training. Why the 8-day NDVI data from Landsat 8 is not used? Its spatial resolution is more suitable for urban study. Furthermore, its temporal resolution is consistent with other geographic products.

5.      In Table 5, the RMSE of Kenley Airfield is 1.055, which is more than 1 K. This significant error requires further explanation in the paper.

6.      Similarly, in Fig.6, the temporal variations of hourly air temperature in regression results and the Kenley Airfield station have different comparison results with the other weather stations. Its air temperature is lower than the estimated results at the peak point. The reason for this difference should be illustrated.

Reviewer 3 Report

This is a well-written paper containing interesting results, and the submission is worth of publication. Following are some minor comments:

1. It is recommended that the focus of the abstract be on the description and possible application of the research results, the current abstract is not concise enough.

2. The keyword should appear in the abstract, but the keyword "data fusion" is not found in the abstract, please check it carefully again.

3. Line 32 mentions UNDRR for the first time. Please complete the full name of UNDRR.

4. Countries with different latitudes and different climate zones have different definitions of the temperature range of heat waves. It is recommended to add the definition of heat waves in the introduction.

5. The citations in the article are particularly irregular, such as lines 52, 78, and 79. There are three or more authors. Why only list two authors instead of "the first author + et al."?

6. Lines 37-38 of the introduction mention that "heatwaves pose a significant risk to these rapidly urbanising regions, especially in the big cities of developing countries". Why did London, a developed region, be chosen as the research area?

7. It is suggested to add the scientific question of this study in the introduction section.

8. The research paper is not a graduation thesis, it is recommended to delete lines 125-130.

9. It is recommended to add the implications of this study in the conclusion section.

Reviewer 4 Report

Comments to the Authors

The study proposed a two-step data fusion model to generate temporally continuous hourly air temperature datasets during heat waves. The authors demonstrate the generated results incorporating multi-source data and the statistical downscaling model. The study was comprehensive and the results seem reasonable. The manuscript is in general well written and brings out the merit of the proposed method. It may be recommended for acceptance post satisfactory response to some of my concerns. Some comments are shown below:

 Major Comments

1. Introduction provides insufficient references. More papers should be cited in the reference, such as:

[1] Jin, Z., Ma, Y., Chu, L., Liu, Y., Dubrow, R., Chen, K., 2022. Predicting spatiotemporally-resolved mean air temperature over Sweden from satellite data using an ensemble model. Environmental Research 204, 111960. https://doi.org/10.1016/j.envres.2021.111960

[2] Li, X., Zhou, Y., Asrar, G.R., Zhu, Z., 2018. Developing a 1 km resolution daily air temperature dataset for urban and surrounding areas in the conterminous United States. Remote Sensing of Environment. https://doi.org/10.1016/j.rse.2018.05.034

[3] Liu, Y., Zhang, X., Gao, Y., Qu, T., Shi, Y., 2022. Improved CNN Method for Crop Pest Identification Based on Transfer Learning. Computational Intelligence and Neuroscience 2022. https://doi.org/10.1155/2022/9709648

[4] Shen, H., Jiang, Y., Li, T., Cheng, Q., Zeng, C., Zhang, L., 2020. Deep learning-based air temperature mapping by fusing remote sensing, station, simulation and socioeconomic data. Remote Sensing of Environment 240, 111692. https://doi.org/10.1016/j.rse.2020.111692

[5] Zhu, X., Zhang, Q., Xu, C.Y., Sun, P., Hu, P., 2019. Reconstruction of high spatial resolution surface air temperature data across China: A new geo-intelligent multisource data-based machine learning technique. Science of the Total Environment 665, 300–313. ttps://doi.org/10.1016/j.scitotenv.2019.02.077

2.  Section 4 “Results and discussions” should be divided and more detailed. 

 Minor Comments

1. Figure 1- Figure 3, high-resolution images were required.

Round 2

Reviewer 2 Report

Precisely because the research objective of this paper is to propose a reliable method to obtain urban temperature data with high spatiotemporal resolution. It is very important to verify the reliability of the two-step method. At present, the modification has not been fully demonstrated, and more validation and comparative experiments with other models are needed to demonstrate the reliability and advantages of the proposed method.

The writing is easy to understand.

Reviewer 3 Report

Authors have provided good revisions and feedbacks to all reviewers' comments. All my questions are resolved.

Author Response

Dear reviewer:

Thank you for recognizing our research! Your previous suggestions have greatly benefited our study.

Thanks again!